🔓 | **Open Peer Review** | *Clinical Microbiology* | *Research Article*

# A novel *TRIM22* gene polymorphism promotes the response to PegIFNα therapy through cytokine-cytokine receptor interaction signaling pathway in chronic hepatitis B

Long Wang,[1,2] Ni Lin,[2] Yanfang Zhang,[1,2] Shaoying Guo,[2] Can Liu,[1,2,3] Caorui Lin,[1,3] Yongbin Zeng,[1,3] Wennan Wu,[1,3] Jianhui Guo,[1,3] Chenggong Zhu,[1,2] Fuguo Zhan,[1,3] Qishui Ou,[1,2,3] Zhen Xun[1,2,3]

**ABSTRACT** Objectives: Pegylated interferon alfa (PegIFNα) has limited efficacy in patients with chronic hepatitis B (CHB). Because single-nucleotide polymorphisms (SNPs) are known to confer disease susceptibility and influence treatment response, we aimed to identify novel tripartite motif-containing 22 (TRIM22) SNPs that were associated with therapeutic efficacy in patients with CHB after PegIFNα treatment. Samples from 107 patients with CHB were genotyped using Asian Screening Array gene chips. The related mechanisms of SNPs screened were explored through both cell experiments and RNA sequence methods. *TRIM22* was the most upregulated upon stimulation with PegIFNα. Specifically, the SNP rs10838543 CC genotype in *TRIM22* was found to be associated with the positive response to PegIFNα treatment. The SNP rs10838543 CC genotype in *TRIM22* was more stable and more robustly inhibited hepatitis B virus (HBV) replication in HepAD38 cells compared to the TT genotype. Mechanistically, we showed that the cytokine-cytokine receptor interaction signaling pathway was significantly upregulated in HepAD38 cells stably expressing the SNP rs10838543 CC genotype of *TRIM22* compared to the TT genotype after PegIFNα treatment and that the SNP rs10838543 CC genotype in *TRIM22* enhanced PegIFNα-induced anti-HBV activity by inducing the secretion of IFNL1, CCL3, and CCL5. The SNP rs10838543 CC genotype in *TRIM22* increased the secretion of the cytokines IFNL1, CCL3, and CCL5 from hepatocytes by regulating the cytokine-cytokine receptor interaction signaling pathway and was positively correlated with the PegIFNα-induced treatment response in patients with CHB. Our findings suggest that genotyping patients with CHB for the SNP rs10838543 in *TRIM22* may be a useful biomarker to help physicians identify patients who are most likely to benefit from PegIFNα treatment.

**IMPORTANCE** Pegylated interferon alfa (PegIFNα) has limited efficacy in the treatment of chronic hepatitis B (CHB). Although many biomarkers related to hepatitis B virus (HBV) have been proposed to stratify patients, the response rate to PegIFNα is still unsatisfactory. Herein, our data suggest that the single-nucleotide polymorphism (SNP) rs10838543 in TRIM22 potentiates a positive clinical response to PegIFNα treatment in patients with hepatitis B e antigen-positive CHB by increasing the levels of IFNL1, CCL3, and CCL5. These observations can help guide treatment decisions for patients with CHB to improve the response rate to PegIFNα.

**KEYWORDS** tripartite motif-containing 22, chronic hepatitis B, pegylated interferon alfa, single-nucleotide polymorphism

H epatitis B virus (HBV) is a major global health issue, and it contributes to increasing rates of liver cirrhosis and/or hepatocellular carcinoma (HCC) (1). According to the

Address correspondence to Qishui Ou, ouqishui@fjmu.edu.cn, or Zhen Xun, xunzhen@fjmu.edu.cn.

Long Wang, Ni Lin, and Yanfang Zhang contributed equally to this article. Author order was determined by drawing straws.

The authors declare no conflict of interest.

See the funding table on p. 11.

World Health Organization (WHO), approximately 257 million people carry hepatitis B surface antigen (HBsAg); it is endemic in many countries, with higher prevalence in Asia and parts of Africa compared with North America and Europe (2–4). HBV is usually treated with pegylated interferon alfa (PegIFNα) or with nucleos(t)ide analogs (NAs) (5). PegIFNα treatment has advantages over NAs, including higher rates of hepatitis B e antigen (HBeAg) loss and seroconversion as well as sustained virological responses and HBsAg loss after treatment. Patients are also less likely to develop resistance to PegIFNα. Nevertheless, some patients with HBV cannot be treated with PegIFNα owing to negative side effects or failure to respond to treatment (6–8), and biomarkers to predict which patients with chronic hepatitis B (CHB) are likely to respond favorably to PegIFNα therapy are urgently needed.

Currently, the stratification of patients to receive PegIFNα therapy is mainly based on the the presence of virological indicators related to HBV, such as presence of HBsAg, HBeAg, and HBV DNA (9). Some studies have also proposed that, in patients with CHB, the level of hepatitis B core antibody (anti-HBc) and/or HBV RNA may predict a positive treatment response to PegIFNα therapy (10, 11). However, current clinical practice sees only about one-third of patients with CHB responding to PegIFNα treatment (2). Although it is known that reciprocal interactions between HBV and the host contribute substantially to the efficacy of antiviral drugs, there have been relatively few studies to identify host genetic markers with potential prognostic value regarding whether patients may benefit from antiviral therapies (12, 13). Single-nucleotide polymorphisms (SNPs) have been associated with individual disease susceptibility and differential treatment responses for many diseases (14), and the *signal transducing activator of transcription 4* (*STAT4*) rs7574865 allele is a reliable predictor of response to PegIFNα therapy in patients who are HBeAg positive (15). In addition, *interferon-induced protein with tetratricopeptide repeats 1* (*IFIT1*) regulates the patient response to PegIFNα treatment in patients with CHB, and its rs303218 polymorphism can predict endpoint virological responses (16). Although these polymorphisms have been associated with patient response to PegIFNα, so far, studies have not determined the molecular mechanisms that result in these particular alleles conferring sensitivity to PegIFNα among patients with HBV.

The tripartite motif (TRIM) gene family and its encoded proteins are involved in various cellular processes, including cell proliferation, differentiation, development, oncogenesis, and apoptosis (17, 18). TRIM proteins are also known as RING-B-box-coiled-coil (RBCC) proteins because they contain an RBCC motif, which consists of a really interesting new gene (RING) domain, one or two B-boxes, and a predicted coiled-coil region. Since the first RBCC motif was discovered in Xenopus nuclear factor 7 (XNF7), more than 80 members of this family have been found in humans (19). Evidence has linked TRIM proteins to viral infection, and studies by Stremlau et al. have shown that TRIM5-α has a strong inhibitory effect on the replication of human immunodeficiency virus 1 (HIV-1) (20, 21). Another family member, TRIM31, regulates K63-linked polyubiquitination of mitochondria-associated membranes (MAVs), thereby regulating the aggregation of MAVs (22). *In vitro*, TRIM22 has been found to play a role in the response of patients with hepatitis C virus (HCV) infection to PegIFNα treatment. Specifically, PegIFNα treatment induces TRIM22-mediated ubiquitination of nonstructural 5A (NS5A) protein, which is essential for HCV replication, leading to NS5A degradation and preventing viral replication (23). TRIM22 has also been shown to have anti-HBV activity and is associated with the efficacy of PegIFNα (24, 25), but whether *TRIM22* polymorphisms influence the response of patients with CHB to treatment with PegIFNα is not clear.

In this study, we discovered that the SNP rs10838543 genotype in *TRIM22* was associated with the response to PegIFNα therapy in 107 CHB patients. Furthermore, we found that the SNP rs10838543 CC genotype in *TRIM22* increased secretion of the cytokines IFNL1, CCL3, and CCL5 from hepatocytes by regulating the cytokine-cytokine receptor interaction signaling pathway and was positively correlated with the PegIFNα-induced treatment response in patients with HBeAg-positive CHB. Our findings identify

a novel genetic biomarker associated with treatment response in patients with CHB treated with PegIFNα and suggest that genotyping patients with CHB for the SNP rs10838543 in *TRIM22* may be warranted to guide therapy selection.

## MATERIALS AND METHODS

### Patients and healthy participants

Patients with CHB who were treated with PegIFNα at the First Affiliated Hospital of Fujian Medical University and for whom peripheral blood samples were available were identified and considered for inclusion in the study. The inclusion criteria were as follows: (i) regularly received PegIFNα treatment for 48 weeks (including PegIFNα monotherapy, PegIFNα, and NAs as combination or sequential therapy; hereafter, all of these treatment regimens are collectively referred to as PegIFNα treatment); (ii) age >16 years; (iii) HBsAg positive for ≥6 months; (iv) HBeAg positive; (v) HBV DNA $\geq 2 \times 10^4$ IU/mL; (vi) serum alanine aminotransferase (ALT) ≥upper limit of normal; and (vii) no previous antiviral treatment (3). Patients with concomitant infection with HCV, hepatitis delta virus, HIV, or any other viral infectious disease were excluded. Patients included in the study were divided into two groups: complete response (CR) and suboptimal response (SR) to PegIFNα treatment. Patients were included in the CR group if they had HBV DNA <500 IU/mL and were HBeAg negative or demonstrated serological conversion and/or HBsAg clearance. Patients were included in the SR group if they had HBeAg >1 COI or HBV DNA >2,000 IU/mL or ALT >upper limit of normal at the 24 weeks post-treatment follow-up visit (week 72) (2, 5, 26). Sex and serological markers before PegIFNα treatment are shown in Table S1.

### SNP genotyping

Peripheral blood samples for genotyping had been collected at various times during the treatment period and stored in anticoagulative tubes. The samples were genotyped using Asian Screening Array (ASA) chips at the BioMiao Biological Technology (Beijing) Co., Ltd following the manufacturer's protocol.

### Polymerase chain reaction

Quantitative real-time polymerase chain reaction (qRT-PCR) was performed to quantify mRNA levels. Total RNA was extracted from cultured cells using TRIzol reagent (Invitrogen, USA). RNA was reverse-transcribed to cDNA using the cDNA Synthesis Kit (Thermo Fisher, St. Louis, MO, USA), according to the manufacturer's protocol. qRT-PCR was performed on a Duant Studio Dx (Applied Biosystems, USA) using SYBR Green qPCR Master Mixes (Vazyme Biotech Co., Ltd). mRNA expression was normalized to glyceraldehyde 3-phosphate dehydrogenase and calculated using the $2^{-\triangle Ct}$ method. The primer sequences used in this study are shown in Table S2.

### Cell culture

Dulbecco's modified Eagle's medium (DMEM) was purchased from Hyclone (Logan, UT, USA). Fetal bovine serum (FBS) and 0.25% trypsin-ethylenediaminetetraacetic acid were obtained from Gibco (Detroit, MI, USA). Penicillin-streptomycin (PS) was purchased from Beyotime Biotechnology (Shanghai, China). Human HCC cell lines were obtained from the Cell Bank of the Typical Culture Preservation Committee of the Chinese Academy of Sciences and included HepG2, HepG2.2.15, Huh7, and HepAD38. Cells were maintained in DMEM containing 10% FBS and 1% PS in a 37°C humidified incubator with 5% $CO_2$ saturation.

### Lentivirus transfection

Lentiviral vectors expressing the different *TRIM22* rs10838543 genotypes were purchased from Hanheng Biotechnology (Shanghai, China). Cells were seeded in six-well plates at a

density of $1 \times 10^5$ cells per well for 24 hours before transfection. Twenty-four hours after transfection, the serum‑free medium was replaced with a fresh medium containing 10% serum. Forty-eight hours after transfection, puromycin was added to select cells stably expressing the vector. The transfection efficiency was detected by fluorescence microscopy and flow cytometry.

## Stability detection of TRIM22 mRNA

Cells were seeded in six-well plates at a density of $2 \times 10^5$ cells per well for 24 hours, then cells were collected at 0, 0.5, 1, 1.5, 2, and 3 hours after actinomycin D treatment, and RNA was extracted. The mRNA level of *TRIM22* was detected by qRT-PCR, and the data were analyzed to calculate the half-life of different genotypes of *TRIM22*.

## Western blot analysis

Cells were lysed in cell lysis buffer containing protease inhibitors (Beyotime, Nanjing, Jiangsu, China). After centrifugation at 12,000× *g* for 5 min, the supernatant was collected. Then, protein samples were loaded on 10% or 8% of polyacrylamide gels and separated by sodium dodecyl sulfate-polyacrylamide gel electrophoresis (SDS-PAGE). Target proteins were transferred onto nitrocellulose membranes (Beyotime, Nanjing, Jiangsu, China). Subsequently, the membranes were blocked with 5% nonfat milk for 1 hour at room temperature. Anti-TRIM22 antibodies (Atlas Antibodies Cat# HPA003575, RRID: AB_1080378) were diluted to appropriate concentrations in the blocking solution and incubated with the membranes overnight at 4°C. After incubation with the horseradish peroxidase (HRP)-conjugated secondary antibody (Beyotime, Nanjing, Jiangsu, China) for 1 hour at 37°C, the proteins were detected and visualized using the gel imaging analyzer (Beijing LiuYi Biotechnology WD-9413B, China).

## Statistical analysis

Data are expressed as the mean ± SD unless indicated otherwise. Data were analyzed with Prism 8 (GraphPad) software using the Mann-Whitney *U* test or unpaired Student's *t*-test when comparing two independent groups, and the $\chi^2$ test to examine differences in categorical variables. Data from a representative experiment of two or three independent replicates are shown. All *P* values were two-tailed and statistical significance was defined as $P < 0.05$.

## RESULTS

### PegIFNα treatment upregulates TRIM22 expression

To identify TRIM genes associated with the treatment response to PegIFNα treatment, we treated HepAD38 and HepG2.2.15 cells, which are hepatoblastoma cells stably transfected to express HBV, with 1,000 IU/mL PegIFNα or with phosphate-buffered saline (PBS) for 24 hours (the cell lines contain 1.1 times the HBV genome and two times the HBV genome linked head to tail, respectively). qRT-PCR analysis of cell lysates showed that *TRIM22* mRNA was statistically significantly higher in PegIFNα-treated cells compared to PBS-treated cells ($P < 0.001$ for both cell lines). *TRIM14* expression was also statistically significantly higher in PegIFNα-treated cells ($P = 0.0186$ and $P = 0.0153$ in HepAD38 and HepG2.2.15 cells, respectively), and *TRIM38* mRNA was statistically significantly higher in HepG2.2.15 cells ($P = 0.0141$). The mRNA levels of other TRIM genes were similar between treatment groups (Fig. 1A and B). We next evaluated TRIM22 protein levels by western analysis, and we confirmed that PegIFNα treatment increased the amount of TRIM22 protein compared to control-treated cells in both cell lines (Fig. 1C and D). These results demonstrate that PegIFNα stimulates TRIM22 expression *in vitro*.

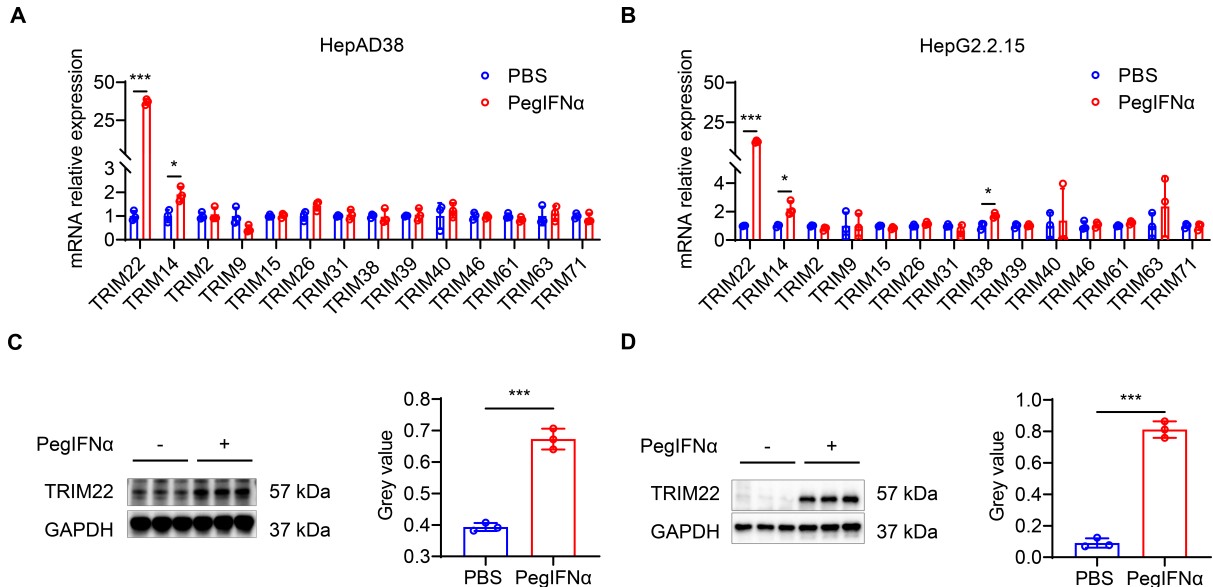

**FIG 1** PegIFNα treatment upregulates TRIM22 expression. HepAD38 and HepG2.2.15 cells were plated at a medium density of $1 \times 10^5$ cells and, after attachment, treated with 1,000 IU/mL PegIFNα or with sterile PBS for 24 hours. (A and B) mRNA level of TRIM-family genes determined by qRT-PCR in (A) HepAD38 and (B) HepG2.2.15 cells. (C and D) (Left) Western blot and (right) quantification of TRIM22 protein in (C) HepAD38 and (D) HepG2.2.15 cells. Data from a representative experiment of two or three independent replicates are shown. Unpaired Student's t-test, *P < 0.05, **P < 0.01, ***P < 0.001.

## SNP rs10838543 CC genotype in TRIM22 is associated with PegIFNα response

Based on findings from ASA genotyping (data were not shown) and the aforementioned characterization of cell line models, we prioritized two SNPs for further analysis: *TRIM22* rs10838543 and rs7112085. We showed that *TRIM22* rs10838543 was statistically significantly distributed between the CR and SR groups (OR = 2.68, Table 1). Also, under the dominant model, the frequencies of genotype CT and CC were higher in the CR group compared to the SR group (OR = 3.16, Table 1), and allele C was higher in the CR group compared to the SR group (OR = 3.09, Table 1). The distributions of allele frequency and genotype frequency for *TRIM22* rs7112085 were not associated with

**TABLE 1** Relationship between genotypes and allele frequencies of TRIM SNPs between CR and SR groups[b]

| SNP | Genotype | SR (n = 77) | CR (n = 30) | P | OR | 95% CI |
|---|---|---|---|---|---|---|
| TRIM22 rs10838543 | | | | | | |
| Allele | T | 139 (90.3) | 45 (75.0) | 0.004[a] | 3.089 | 1.456–6.530 |
| | C | 15 (9.7) | 15 (25.0) | | | |
| Genotype | TT | 62 (80.5) | 17 (56.7) | | | |
| | CT | 15 (19.5) | 11 (36.7) | 0.038[a] | 2.675 | 1.006–6.798 |
| | CC | 0 (0) | 2 (6.6) | 0.053 | - | - |
| Dominate | TT vs CT + CC | 62/15 | 17/13 | 0.012[a] | 3.161 | 1.301–7.694 |
| Recessive | CC vs CT + TT | 0/77 | 2/28 | 0.077 | 0 | 0.000-0.828 |
| TRIM22 rs7112085 | | | | | | |
| Allele | A | 105 (68.2) | 44 (73.3) | 0.462 | 0.779 | 0.411–1.490 |
| | G | 49 (31.8) | 16 (23.7) | | | |
| Genotype | AA | 37 (48.1) | 14 (46.7) | | | |
| | AG | 31 (40.3) | 16 (53.3) | 0.479 | 1.364 | 0.570–3.368 |
| | GG | 9 (11.7) | 0 (0.0) | 0.1 | 0 | 0.000–1.275 |
| Dominate | AA vs AG + GG | 37/40 | 14/16 | 0.898 | 1.057 | 0.456–2.521 |
| Recessive | GG vs AG + AA | 9/68 | 0/30 | 0.059 | - | - |

[a]P < 0.05.
[b]CR, complete response; SR, suboptimal response.

PegIFNα response (Table 1). These results suggest that the rs10838543 CC genotype in *TRIM22* may contribute to the response of patients with CHB to PegIFNα treatment.

## SNP rs10838543 CC genotype in TRIM22 has a higher TRIM22 mRNA level and is more stable than the TT genotype

All patients harboring the rs10838543 SNP ($n$ = 100) were divided into two groups based on the genotype of the *TRIM22* locus: the TT group and the (CT + CC) group. Patients with the rs10838543 (CT + CC) genotype had higher *TRIM22* expression compared to patients with the rs10838543 TT genotype ($P$ = 0.0133) (Fig. 2A). Next, we quantified the endogenous expression of *TRIM22* in different cell lines, including HepG2, HepG2.2.15, Huh7, and HepAD38. HepAD38 cells had the lowest level of *TRIM22* mRNA (Fig. 2B) and were used to test the effects of different patient *TRIM22* rs10838543 genotypes. Lentiviral vectors were constructed to stably express the SNP rs10838543 CC genotype in *TRIM22* (Lv-TRIM22-CC) or the SNP rs10838543 TT genotype in *TRIM22* (Lv-TRIM22-TT) and transfected into HepAD38 cells. Expression of the vector was confirmed using qRT-PCR and western analysis (Fig. 2C; Fig. S1). In RNAfold analysis, SNP rs10838543 changed the centroid secondary structure and changed the minimum free energy (MFE) of the RNA molecules from −273.60 kcal/mol (rs10838543 TT genotype) to −326.50 kcal/mol

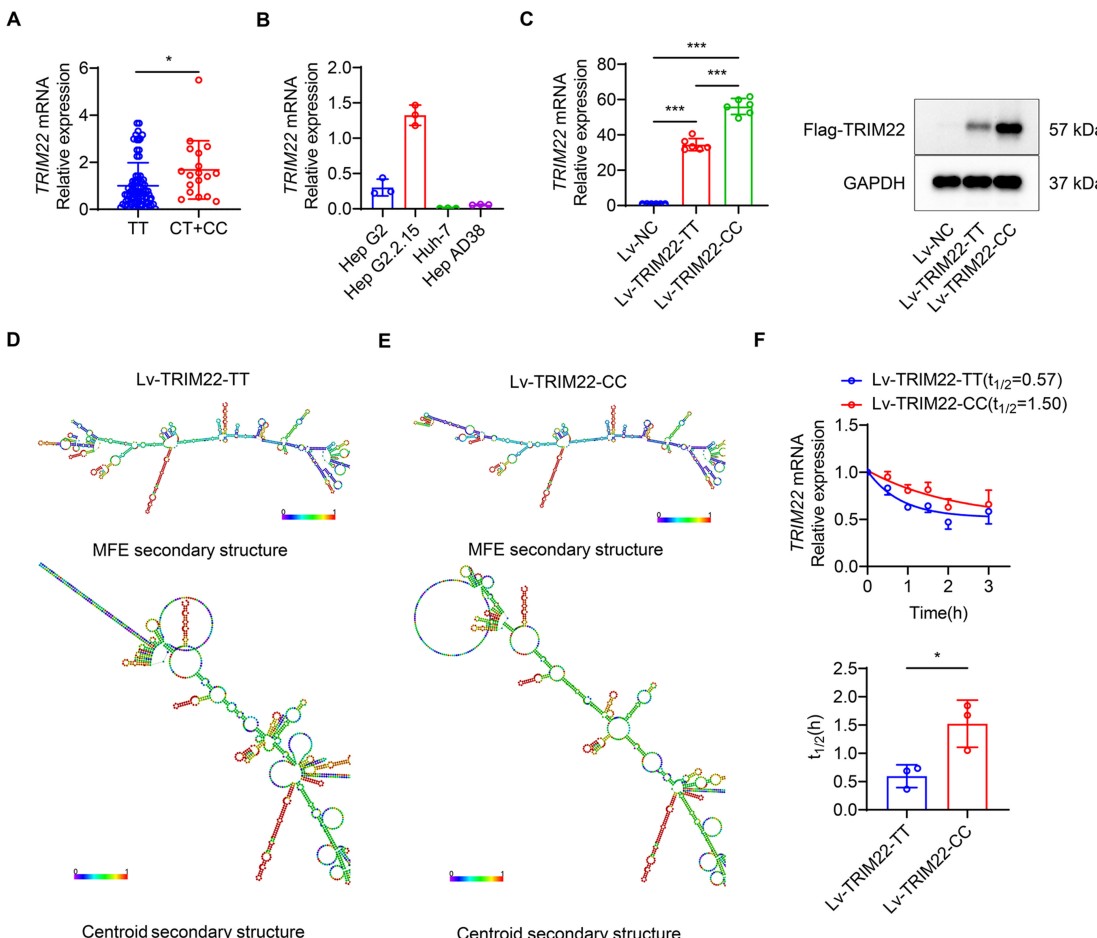

**FIG 2** The SNP rs10838543 CC genotype in *TRIM22* has a higher *TRIM22* mRNA level and is more stable than the TT genotype. (A) mRNA level of *TRIM22* in patients carrying SNP rs10838543 who were treated with PegIFNα according to genotype. (B) *TRIM22* mRNA level in four different cell lines. (C) Level of TRIM22 expression (left: mRNA by qRT-PCR; right: protein by western analysis). (D) MFE and centroid secondary structure of SNP rs10838543 TT genotype in *TRIM22*. (E) MFE and centroid secondary structure of SNP rs10838543 CC genotype in *TRIM22*. (F) Changes in *TRIM22* mRNA with time after actinomycin D treatment and the half-life of *TRIM22* mRNA of different genotypes. Data from a representative experiment of two or three independent replicates are shown. Unpaired Student's $t$-test, *$P$ < 0.05, **$P$ < 0.01, ***$P$ < 0.001.

(rs10838543 CC genotype) (Fig. 2D and E). Next, we detected the stability of mRNA by actinomycin D, the results showed that the half-life of mRNA of SNP rs10838543 CC genotype in *TRIM22* ($t_{1/2}$=1.50 hours) was significantly longer than the TT genotype ($t_{1/2}$=0.57 hours) (Fig. 2F). These findings indicate that the *TRIM22* mRNA of rs10838543 CC genotype had higher mRNA level and was more stable in structure compared with TT genotype.

## SNP rs10838543 CC genotype in TRIM22 inhibits HBV replication *in vitro*

Then, we measured the expression of HBV genes/proteins, which showed that Lv-TRIM22-CC-infected cells had lower *preS1* mRNA expression, levels of HBsAg and HBeAg in the supernatant compared to Lv-TRIM22-TT-infected cells ($P < 0.001$) (Fig. 3A through D). In addition, Lv-TRIM22-CC-infected cells had significantly higher levels of IL-8 and IFN-γ in the supernatant compared to Lv-TRIM22-TT-infected cells (Fig. 3E). Taken together, we conclude that the SNP rs10838543 CC genotype in *TRIM22* has a more robust inhibitory effect on HBV replication than the SNP rs10838543 TT genotype in *TRIM22 in vitro*.

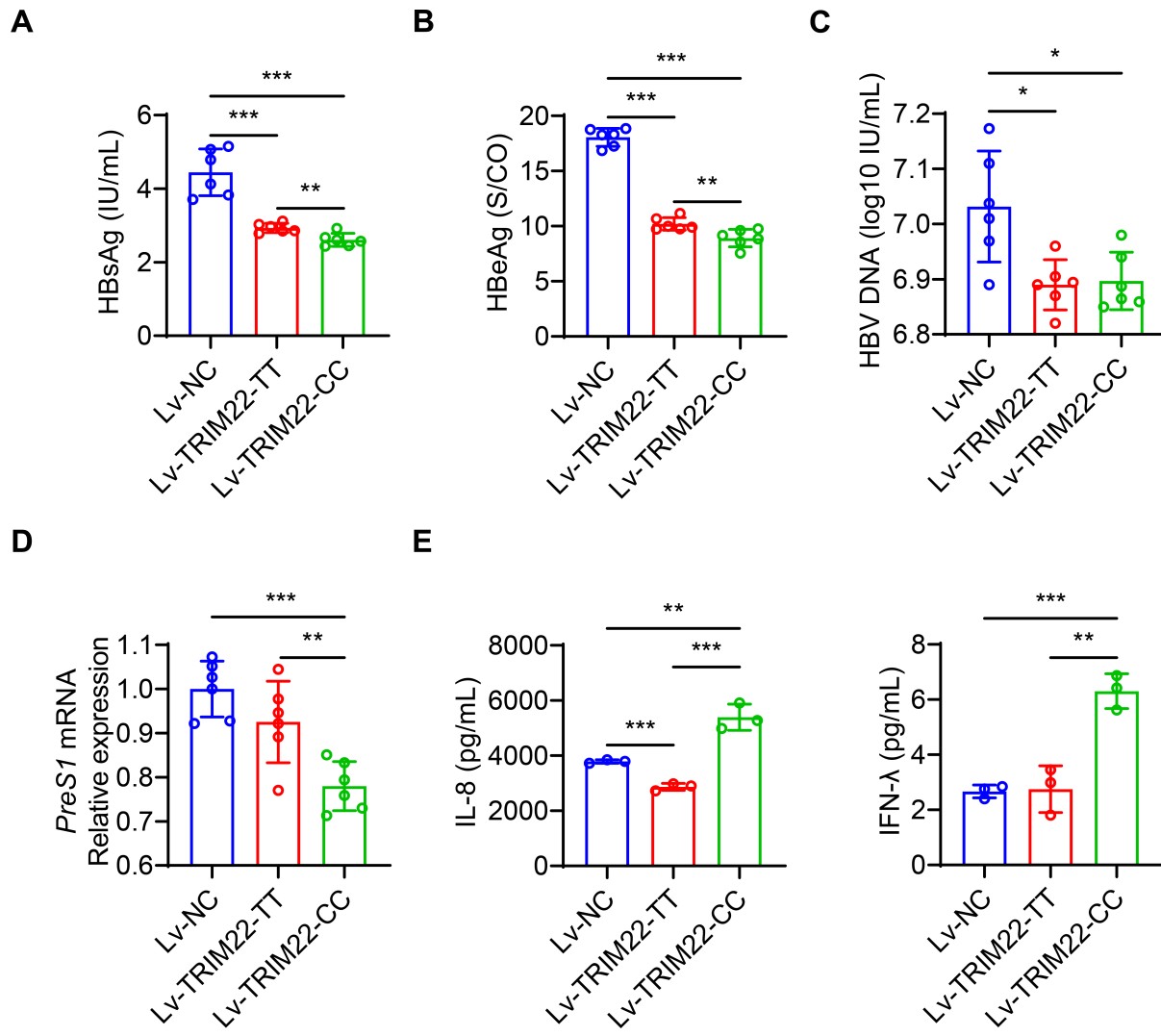

**FIG 3** The SNP rs10838543 CC genotype in *TRIM22* inhibits HBV replication (A–C) HBsAg, HBeAg, and HBV DNA in the culture supernatants and *preS1* mRNA (D) of HepAD38 cells stably expressing *TRIM22* or control vector. (E and F) Levels of indicated cytokines in the culture supernatants of HepAD38 cells stably expressing *TRIM22* or control vector. Data from a representative experiment of two or three independent replicates are shown. Unpaired Student's *t*-test, *$P <$ 0.05, **$P < 0.01$, ***$P < 0.001$.

## SNP rs10838543 in TRIM22 induces the cytokine-cytokine receptor interaction signaling pathway after PegIFNα treatment

PegIFNα can induce various IFN-stimulated genes through JAK-STAT signaling (27). Therefore, we investigated the molecular mechanism by which different genotypes of *TRIM22* may affect the clinical response to PegIFNα by regulating JAK‑STAT signaling. First, we confirmed that the different *TRIM22* rs10838543 expression vectors did not differentially modulate the levels of STAT1, STAT2, and STAT3 protein nor their phosphorylation status (Fig. S2). Then, to determine the underlying mechanisms by which different SNP rs10838543 genotypes in *TRIM22* inhibit HBV, we treated our HepAD38 cell lines stably expressing Lv-TRIM22-CC or Lv-TRIM22-TT with 1,000 IU/mL PegIFNα for 24 hours. RNA-sequencing (RNA-seq) analysis of cell lysates identified genes that were significantly (at least twofold) upregulated or downregulated in PegIFNα-treated cells expressing Lv-TRIM22-CC compared to PegIFNα-treated cells expressing Lv-TRIM22-TT (Fig. 4A), and we confirmed the RNA-seq results using qRT-PCR (Fig. S3). Kyoto Encyclopedia of Genes (KEGG) and Gene Set Enrichment Analysis (GSEA) revealed that the cytokine-cytokine

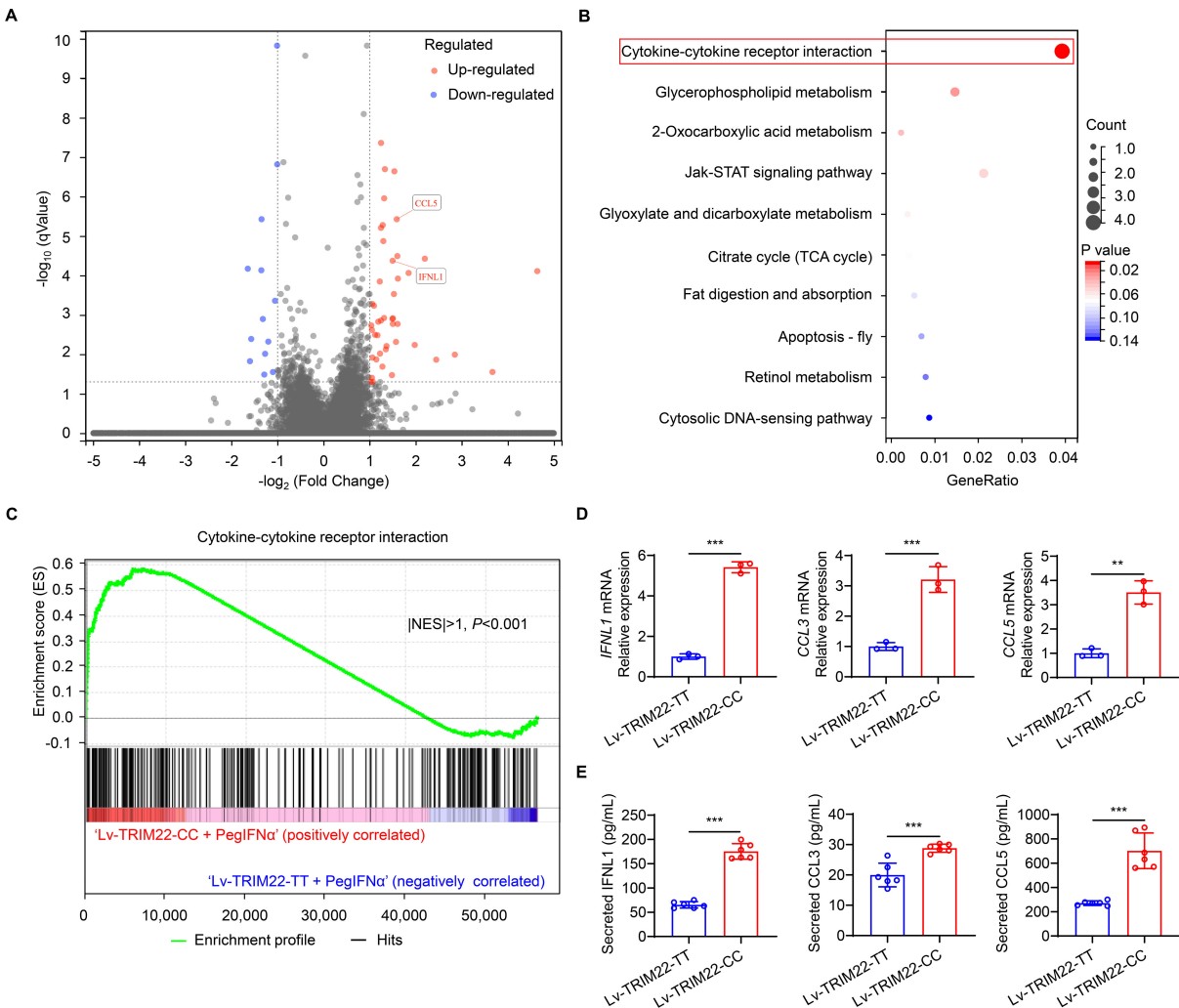

**FIG 4** The SNP rs10838543 genotype in *TRIM22* affects the cytokine-cytokine receptor interaction signaling pathway in PegIFNα-treated cells. (A) Volcano plots showing RNA-seq analysis of HepAD38 cells stably expressing Lv-TRIM-CC compared to Lv-TRIM-TT (*n* = 3); blue: downregulated genes, red: upregulated genes. (B) KEGG pathway enrichment analysis and (C) GSEA analysis of dysregulated genes identify upregulation of the cytokine-cytokine receptor interaction signaling pathway. (D) qRT-PCR analysis of *IFNL1*, *CCL3*, and *CCL5* mRNA and (E) protein levels in HepAD38 cells stably expressing *TRIM22* genotypes as indicated after PegIFNα treatment. Data from a representative experiment of two or three independent replicates are shown. Unpaired Student's *t*-test, \*P < 0.05, \*\*P < 0.01, \*\*\*P < 0.001.

receptor interaction signaling pathway was significantly upregulated in Lv-TRIM22-CC-expressing cells treated with PegIFNα (Fig. 4B and C), and the most-affected genes in this pathway were *IFNL1* (five-fold higher expression), *CCL3* (three-fold higher expression), and *CCL5* (three-fold higher expression) (Fig. 4D). ELISA analysis of cell culture supernatants confirmed that PegIFNα treatment of HepAD38 cells expressing Lv-TRIM22-CC induced higher levels of IFNL1, CCL3, and CCL5 protein compared to Lv-TRIM22-TT (Fig. 4E). We repeated these analyses in the same cell lines without PegIFNα stimulation and found similar patterns regarding differential regulation of genes and cytokines ( Fig. S4). These data indicate that the SNP rs10838543 CC genotype in *TRIM22* increases the secretion of IFNL1, CCL3, and CCL5 and upregulates the cytokine-cytokine receptor interaction signaling pathway and that PegIFNα treatment further stimulates these effects, resulting in improved anti-HBV efficacy.

## DISCUSSION

PegIFNα is the only licensed immunomodulator to treat HBV. Despite virological and serological indicators that have been shown to predict which patients will benefit from PegIFNα treatment, it currently cures less than half of patients with HBeAg-positive CHB (2, 26, 28). Previous reports have shown that several TRIM proteins are modulated by PegIFNα and play a role in restricting viral infection (29–31). However, few studies have investigated the role of SNPs in *TRIM* genes to affect clinical efficacy in patients with CHB who receive PegIFNα therapy. In this study, we demonstrated that a novel *TRIM22* gene polymorphism, SNP rs10838543 in *TRIM22*, promotes the response to PegIFNα therapy by elevating the levels of IFNL1, CCL3, and CCL5 through cytokine-cytokine receptor interaction signaling pathway in CHB patients. Genotyping patients with CHB for the SNP rs10838543 in *TRIM22* may be a useful biomarker to help physicians identify patients who are more likely to respond to PegIFNα therapy.

Studies have shown that genetic factors of the host—in particular, SNPs—can influence the response to antiviral therapy in patients with CHB (32–35). Therefore, in this study, we performed SNP genotyping of peripheral blood from HBeAg-positive patients with CHB treated with PegIFNα using ASA gene array chips, and we found that several SNPs in TRIM genes were associated with clinical benefit from PegIFNα treatment (data were not shown). In HepAD38 and HepG2.2.15 cells *in vitro*, PegIFNα stimulated *TRIM22* and *TRIM14* expression, which was consistent with previous reports (36). Moreover, we showed that the SNP rs10838543 in *TRIM22* was associated with a markedly improved patient response to PegIFNα treatment. The *TRIM22* rs10838543 genotype is associated with chronic persistent HBV infection (37), but the relationship between SNP rs10838543 in *TRIM22* and response to PegIFNα in patients with CHB, which we report in this study, represents a novel potential biomarker to stratify patients to treatment.

TRIM22 inhibits the activity of the HBV core promoter (CP) *via* its nuclear-located RING domain (38), but the inhibitory effect of different SNP rs10838543 genotypes in *TRIM22* on HBV replication is not well understood. Here, we compared the differential molecular phenotypes associated with the SNP rs10838543 CT/CC genotype in *TRIM22* with the TT genotype. In patient samples, we determined that *TRIM22* expression was significantly higher in the context of the CT/CC genotype compared to the TT genotype, which may reflect the ability of the synonymous SNP to affect its mRNA structure and stability (39, 40). This was further confirmed by the RNAfold algorithm, which predicted that the SNP rs10838543 CC genotype in *TRIM22* to have a lower centroid secondary structure and MFE compared to the TT genotype. Similarly, *in vitro*, when the CC and TT SNP rs10838543 genotypes in *TRIM22* were stably expressed in HepAD38 cells, *TRIM22* gene and protein expression were much higher with the CC genotype compared to the TT genotype. Further analysis of the cell culture supernatant revealed the SNP rs10838543 CC genotype in *TRIM22* more robustly inhibited HBV, as evidenced by significantly lower levels of the HBV proteins, HBsAg and HBeAg. In addition, further experiments demonstrated that the levels of IL-8 and IFN-γ in the supernatant of cells expressing SNP rs10838543 CC genotype in *TRIM22* were significantly elevated compared to the TT

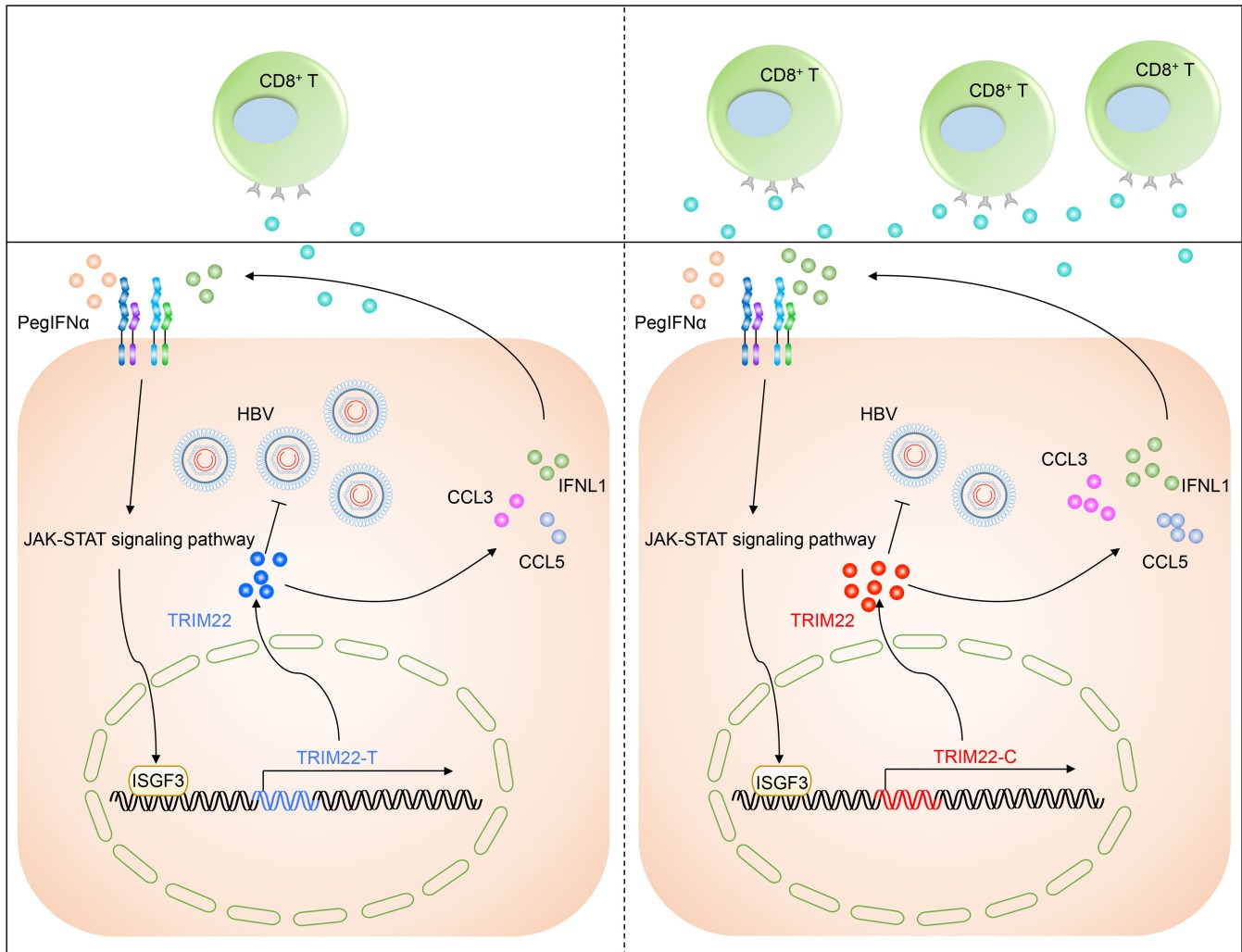

**FIG 5** Proposed model of the effect of the SNP rs10838543 in *TRIM22* on the efficacy of PegIFNα treatment in patients with HBeAg-positive CHB. Compared to patients with CHB with the SNP rs10838543 TT genotype in *TRIM22*, patients with CHB with the CT/CC genotype have higher levels of *TRIM22* mRNA and higher levels of IFNL1, CCL3, CCL5 secretion, resulting in an improved response to PegIFNα treatment.

genotype, which is probably related to the inhibitory effect of the CC genotype on HBV replication.

Current research suggests that PegIFNα inhibits virus replication by inducing the production of antiviral proteins through the JAK-STAT pathway (41, 42). However, we did not observe significant differences in key molecules within the JAK-STAT signaling pathway between cells stably expressing the SNP rs10838543 CC genotype in *TRIM22* compared to the TT genotype, suggesting another mechanism must be responsible for the antiviral effects of this polymorphism. Molecular characterization of the cell line models demonstrated that the cytokine-cytokine receptor interaction pathway was significantly elevated in the context of the SNP rs10838543 CC genotype in *TRIM22* compared to the TT genotype, and IFNL1 was the most significantly overexpressed molecule. Studies have demonstrated that chemokines play an essential role in lymphocyte recruitment during hepatitis progression and that type-III IFN might be essential for the execution of anti-HBV mechanisms after HBV infection (43, 44). Also, IFNL1, which is a type-III IFN family member, may be activated by JAK-STAT signaling, similar to PegIFNα (45). Furthermore, CCL3 and CCL5 have the same receptor, CCR5, which is expressed on various immune cells, including natural killer (NK) cells, NK T cells, CD4-positive T cells, and CD8-positive T cells, and this receptor plays an important role

in HBV infection (41). Consistent with our results, CCL3 and CCL5 levels were increased in cells stably expressing the SNP rs10838543 CC genotype in *TRIM22*, but the specific mechanisms that cause this genotype to be associated with clinical benefit from PegIFNα need to be further explored. One hypothesis is that different SNP rs10838543 genotypes in *TRIM22* differentially affect the cytokine-cytokine receptor interaction pathway, influencing IFNL1 levels. Subsequently, IFNL1 transduces its signal by activating the JAK-STAT signaling pathway, leading to variation in the treatment response among patients treated with PegIFNα. A second hypothesis is that the different SNP rs10838543 genotypes in *TRIM22* may affect the efficacy of PegIFNα by upregulating CCL3 and CCL5, and the crosstalk between CCL3/CCL5 and its receptor, CCR5, may subsequently guide immune cells to infected hepatocytes (43, 46).

There are still some limitations to our current study that will be addressed in further research. First, *TRIM22* is expressed only in human and nonhuman primates, and, owing to a lack of a license for breeding tree shrews, we were unable to validate our findings *in vivo*. Future studies should prioritize controlled *in vivo* studies to confirm the relevance of the SNP rs10838543 genotypes in *TRIM22* for PegIFNα therapy. Second, further delineation of the molecular mechanism that results in differential IFNL expression in the context of different SNP rs10838543 genotypes in *TRIM22,* and whether this might occur due to inhibition of *suppressors of cytokine signaling* (*SOCS*) 1, *SOCS2*, or *SOCS3*, is warranted.

In conclusion, we identified a genetic trait, the SNP rs10838543 CC genotype in *TRIM22*, that is associated with favorable outcomes in patients with CHB treated with PegIFNα. Given the currently low response rate of patients with HBeAg-positive CHB to PegIFNα therapy, our finding that this polymorphism was associated with a positive clinical response in this cohort represents important data to improve patient care. Mechanistically, we demonstrated that this genotype conferred sensitivity to PegIFNα by inducing higher levels of IFNL1, CCL3, and CCL5 (Fig. 5). Thus, our results suggest that screening for the SNP rs10838543 genotype in *TRIM22* may be a useful biomarker to identify patients more likely to benefit from PegIFNα treatment and to improve response rates.

## AUTHOR AFFILIATIONS

[1]Department of Laboratory Medicine, Fujian Key Laboratory of Laboratory Medicine, Gene Diagnosis Research Center, Fujian Clinical Research Center for Clinical Immunology Laboratory Test, The First Affiliated Hospital, Fujian Medical University, Fuzhou, Fujian, China
[2]The First Clinical College, Fujian Medical University, Fuzhou, Fujian, China
[3]Department of Laboratory Medicine, National Regional Medical Center, Binhai Campus of the First Affiliated Hospital, Fujian Medical University, Fuzhou, Fujian, China

## AUTHOR ORCIDs

Qishui Ou http://orcid.org/0000-0002-9923-3212
Zhen Xun http://orcid.org/0000-0002-4242-3402

## FUNDING

| Funder | Grant(s) | Author(s) |
|---|---|---|
| MOST \| National Natural Science Foundation of China (NSFC) | 82030063 | Qishui Ou |
| MOST \| National Natural Science Foundation of China (NSFC) | 82102467 | Zhen Xun |
| MOST \| National Natural Science Foundation of China (NSFC) | 82002217 | Ni Lin |
| Fujian Provincial Health Technology Project | 2020QNA040 | Zhen Xun |

## AUTHOR CONTRIBUTIONS

Long Wang, Validation, Writing – original draft, Writing – review and editing | Ni Lin, Funding acquisition, Writing – original draft | Yanfang Zhang, Data curation | Shaoying Guo, Formal analysis | Can Liu, Data curation, Formal analysis, Funding acquisition, Resources | Caorui Lin, Data curation, Supervision | Yongbin Zeng, Data curation, Methodology | Wennan Wu, Data curation, Formal analysis | Jianhui Guo, Data curation | Chenggong Zhu, Validation | Fuguo Zhan, Investigation, Methodology | Qishui Ou, Data curation, Funding acquisition, Methodology, Resources, Writing – original draft, Writing – review and editing | Zhen Xun, Formal analysis, Funding acquisition, Supervision, Writing – original draft, Writing – review and editing

## ETHICS APPROVAL

This study was conducted in compliance with the 1975 Declaration of Helsinki and was approved by the Ethics Committee of the First Affiliated Hospital of Fujian Medical University (Approval No. MRCTA, ECFAH of FMU [2015]027). Written informed consent was obtained from the study participants.

## ADDITIONAL FILES

The following material is available online.

### Supplemental Material

**Supplemental material (Spectrum02247-23-s0001.docx).** Tables S1 and S2; Fig. S1 to S4.

### Open Peer Review

**PEER REVIEW HISTORY (review-history.pdf).** An accounting of the reviewer comments and feedback.

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
