## [Reviewer comments · Microbiology Spectrum]

Microbiology Spectrum

A novel TRIM22 gene polymorphism promotes the response to PegIFN α therapy through cytokine-cytokine receptor interaction signaling pathway in chronic hepatitis B

Long Wang, Ni Lin, Yanfang Zhang, Shaoying Guo, Can Liu, Caorui Lin, Yong-bin Zeng, Wennan Wu, Jianhui Guo, Chenggong Zhu, Fuguo Zhan, Qishui Ou, and Zhen Xun

Corresponding Author(s): Qishui Ou, Fujian Medical University

Review Timeline:

Submission Date:	May 29, 2023
Editorial Decision:	July 7, 2023
Revision Received:	August 3, 2023
Accepted:	September 8, 2023

Editor: Wei Liu

Reviewer(s): The reviewers have opted to remain anonymous.

Transaction Report:

DOI: <https://doi.org/10.1128/spectrum.02247-23>

July 7, 2023

Mr. Qishui Ou
Fujian Medical University
Fuzhou
China

Re: Spectrum02247-23 (A novel TRIM22 gene polymorphism promotes the response to PegIFN α therapy through cytokine-cytokine receptor interaction signaling pathway in chronic hepatitis B)

Dear Mr. Qishui Ou:

Link Not Available

Sincerely,

Wei Liu

Journals Department
Reviewer comments:

Reviewer #1 (Comments for the Author):

This research entitled "A novel TRIM22 gene polymorphism promotes the response to PegIFN α therapy through cytokine-cytokine receptor interaction signaling pathway in chronic hepatitis B" aimed to find and explain the potential marker for differential PegIFN α response. The study is generally well designed, and demonstrated that the rs10838543 CC genotype in TRIM22 increased secretion of the cytokines IFNL1, CCL3, and CCL5 from hepatocytes by regulating the cytokine-cytokine receptor interaction signaling pathway and was positively correlated with the PegIFN α -induced treatment response in patients with CHB. The study might be interesting for readers since it suggest that genotyping patients with CHB at rs10838543 polymorphism might be a potential biomarker helping physicians identify patients who are most likely to benefit from PegIFN α treatment. But there are still some concerns regarding this study:

1. In Figure 1C right half of the Western blot shows stronger background staining , the author should confirmed that the six lanes with our without PegIFN α treatment were conducted parallely without any manual modification.
2. The authors verify the transfection efficiency of their lentiviral transduction using flow cytometry (Supplemental) but provide no methods for these.
3. Part of the discussion is the repetition of the result. Please clearly pointed out the advantages and shortages of this study.
4. The English writing of the whole text needs to be polished.

Reviewer #2 (Comments for the Author):

Manuscript entitled: "A novel TRIM22 gene polymorphism promotes the response to PegIFN α therapy through cytokine-cytokine receptor interaction signaling pathway in chronic hepatitis B" is well supported and interesting, the authors had experimental evidences in patients and in cell lines, and proved that rs10838543 CC genotype in TRIM22 was found to be associated with positive response to PegIFN α treatment and more robustly inhibited HBV replication in cells compared to the TT genotype. However, there are several concerns that the authors should address in current manuscript.

1. Gene names should be written in italics (in lines 169 and 209, TRIM22 were formatted differently), uniform throughout, please check.
2. Materials and methods should be refined, such as flow cytometry.
3. As far as I know there are more than 80 members of the TRIM family, why these 14 TRIM genes were selected for the study?
4. In Figure 1C, TRIM22 upregulation in the presence of IFN α in HepAD38 cells were not very visible.
5. Although 107 patient samples were used for this study, the results in the tables and figures are not representative of all 107 patients. In the results the authors state (lines 236-239), for example, that 100 people were used. The explanation needs to be given of the particular sampling done for certain analyses.
6. For the RNA fold analysis, the authors showed that a nucleotide change could induce changes in the secondary structure. But the time interval of the actinomycin D test is so long that it basically reaches the plateau after 2 hours, and it is recommended to shorten the time interval.

Staff Comments:

Preparing Revision Guidelines

Please return the manuscript within 60 days; if you cannot complete the modification within this time period, please contact me. If you do not wish to modify the manuscript and prefer to submit it to another journal, please notify me of your decision immediately so that the manuscript may be formally withdrawn from consideration by Microbiology Spectrum.

This research entitled “A novel TRIM22 gene polymorphism promotes the response to PegIFN α therapy through cytokine-cytokine receptor interaction signaling pathway in chronic hepatitis B” aimed to find and explain the potential marker for differential PegIFN α response. The study is generally well designed, and demonstrated that the rs10838543 CC genotype in TRIM22 increased secretion of the cytokines IFNL1, CCL3, and CCL5 from hepatocytes by regulating the cytokine-cytokine receptor interaction signaling pathway and was positively correlated with the PegIFN α -induced treatment response in patients with CHB. The study might be interesting for readers since it suggest that genotyping patients with CHB at rs10838543 polymorphism might be a potential biomarker helping physicians identify patients who are most likely to benefit from PegIFN α treatment. But there are still some concerns regarding this study:

1. In Figure 1C right half of the Western blot shows stronger background staining , the author should confirmed that the six lanes with our without PegIFN α treatment were conducted parallely without any manual modification.
2. The authors verify the transfection efficiency of their lentiviral transduction using flow cytometry (Supplemental) but provide no methods for these.
3. Part of the discussion is the repetition of the result. Please clearly pointed out the advantages and shortages of this study.
4. The English writing of the whole text needs to be polished.

Manuscript entitled: “A novel TRIM22 gene polymorphism promotes the response to PegIFN α therapy through cytokine-cytokine receptor interaction signaling pathway in chronic hepatitis B” is well supported and interesting, the authors had experimental evidences in patients and in cell lines, and proved that rs10838543 CC genotype in TRIM22 was found to be associated with positive response to PegIFN α treatment and more robustly inhibited HBV replication in cells compared to the TT genotype.

However, there are several concerns that the authors should address in current manuscript.

1. Gene names should be written in italics (in lines 169 and 209, TRIM22 were formatted differently), uniform throughout, please check.

2. Materials and methods should be refined, such as flow cytometry.

3. As far as I know there are more than 80 members of the TRIM family, why these 14 TRIM genes were selected for the study?

4. In Figure 1C, TRIM22 upregulation in the presence of IFN α in HepAD38 cells were not very visible.

5. Although 107 patient samples were used for this study, the results in the tables and figures are not representative of all 107 patients. In the results the authors state (lines 236-239), for example, that 100 people were used. The explanation needs to be given of the particular sampling done for certain analyses.

6. For the RNA fold analysis, the authors showed that a nucleotide change could induce changes in the secondary structure. But the time interval of the actinomycin D test is so long that it basically reaches the plateau after 2 hours, and it is recommended to shorten the time interval.

Point-by-point Responses to the Editor and Reviewers' Comments (the Editor and Reviewers' comments are in **bold**):

Major changes to the text in the revised manuscript have been highlighted in yellow.

Reviewer #1: This research entitled "A novel TRIM22 gene polymorphism promotes the response to PegIFN α therapy through cytokine-cytokine receptor interaction signaling pathway in chronic hepatitis B" aimed to find and explain the potential marker for differential PegIFN α response. The study is generally well designed, and demonstrated that the rs10838543 CC genotype in TRIM22 increased secretion of the cytokines IFNL1, CCL3, and CCL5 from hepatocytes by regulating the cytokine-cytokine receptor interaction signaling pathway and was positively correlated with the PegIFN α -induced treatment response in patients with CHB. The study might be interesting for readers since it suggest that genotyping patients with CHB at rs10838543 polymorphism might be a potential biomarker helping physicians identify patients who are most likely to benefit from PegIFN α treatment. But there are still some concerns regarding this study:

1. In Figure 1C right half of the Western blot shows stronger background staining, the author should confirmed that the six lanes with our without PegIFN α treatment were conducted parallely without any manual modification.

Response: We apologize for the confusion. In the revised manuscript, we have reextracted new proteins and performed Western blot to detect the expression level of TRIM22 with or without PegIFN α treatment. The results showed that PegIFN α significantly promoted the expression of TRIM22 protein in HepAD38 cells (**Fig. 1C, as shown below**)

C

Fig. 1. PegIFN α treatment upregulates TRIM22 expression

2. The authors verify the transfection efficiency of their lentiviral transduction using flow cytometry (Supplemental) but provide no methods for these.

Response: Thank you for your comment. In the revised manuscript, we have provided the methods in the supplemental material as follow:

Transfection efficiency of the lentiviral transduction

Evaluation of transduction efficiency was conducted using green fluorescent protein (GFP) expressed in all lentiviruses. Cells were washed three times with PBS, and then dissociated with trypsin. After cell suspensions by DMEM, cells were transferred to suitable FACS tubes. A negative control was performed using untransfected HepAD38 cells. FITC fluorescence was analyzed by flow cytometry and expressed as a percentage. (Supplemental Material, methods, page 3, lines 45-50)

3. Part of the discussion is the repetition of the result. Please clearly pointed out the advantages and shortages of this study.

Response: Thank you for your constructive suggestion. In the revised manuscript, we have added the following text to discuss the advantages and shortages of this study:

In this study, we demonstrated that a novel *TRIM22* gene polymorphism, SNP rs10838543 in *TRIM22*, promotes the response to PegIFN α therapy by elevating levels of IFNL1, CCL3, and CCL5 through cytokine-cytokine receptor interaction signaling pathway in CHB patients. Genotyping patients with CHB for the SNP

rs10838543 in *TRIM22* may be a useful biomarker to help physicians identify patients who are more likely to respond to PegIFN α therapy. (Discussion, pages 15, lines 312-317)

There are still some limitations to our current study that will be addressed in further research. First, *TRIM22* is expressed only in human and nonhuman primates, and, owing to a lack of a license for breeding tree shrews, we were unable to validate our findings *in vivo*. Future studies should prioritize controlled *in vivo* studies to confirm the relevance of the SNP rs10838543 genotypes in *TRIM22* for PegIFN α therapy. Second, further delineation of the molecular mechanism that results in differential IFNL expression in the context of different SNP rs10838543 genotypes in *TRIM22*, and whether this might occur due to inhibition of *suppressors of cytokine signaling* (*SOCS*) 1, *SOCS2*, or *SOCS3*, is warranted. (Discussion, page 18, lines 377-378, line 382)

4. The English writing of the whole text needs to be polished.

Response: We apologize for the language problems in the original manuscript. In the revised manuscript, we have improved the language presentation with the assistance from a native English speaker with appropriate research background.

Reviewer #2: Manuscript entitled: "A novel TRIM22 gene polymorphism promotes the response to PegIFN α therapy through cytokine-cytokine receptor interaction signaling pathway in chronic hepatitis B" is well supported and interesting, the authors had experimental evidences in patients and in cell lines, and proved that rs10838543 CC genotype in TRIM22 was found to be associated with positive response to PegIFN α treatment and more robustly inhibited HBV replication in cells compared to the TT genotype.

However, there are several concerns that the authors should address in current manuscript.

1. Gene names should be written in italics (in lines 169 and 209, TRIM22 were formatted differently), uniform throughout, please check.

Response: Thank you for your suggestion. In the revised manuscript, we have scrutinized the full text and made the appropriate formatting changes. **(Line 3, line 7, line 47, line 118, line 132, line 178, line 189, line 226, lines 261-262, line 284, lines 287-288, line 292, lines 296-297, line 326, line 360, line 362, and line 539)**

2. Materials and methods should be refined, such as flow cytometry.

Response: Thank you for your important comment. In the revised manuscript, we have supplemented the methods for flow cytometry in the Supplemental material as follow:

Transfection efficiency of the lentiviral transduction

Evaluation of transduction efficiency was conducted using green fluorescent protein (GFP) expressed in all lentiviruses. Cells were washed three times with PBS, and then dissociated with trypsin. After cell suspensions by DMEM, cells were transferred to suitable FACS tubes. A negative control was performed using untransfected HepAD38 cells. FITC fluorescence was analyzed by flow cytometry and expressed as a percentage. **(Supplemental Material, methods, page 3, lines 45-50)**

3. As far as I know there are more than 80 members of the TRIM family, why these 14 TRIM genes were selected for the study?

Response: This is an important question. There were indeed more than 80 members of the TRIM families (DOI: 10.1016/j.tibs.2017.01.002). However, the aim of this study was to find genes and their polymorphisms associated with PegIFN α efficacy. In our previous work, we screened only 14 TRIM genes with differences between the CR and SR groups by ASA microarray. Therefore, we selected these 14 TRIM genes for the study.

4. In Figure 1C, TRIM22 upregulation in the presence of IFN α in HepAD38 cells were not very visible.

Response: Thank you for your important comments. In the revised manuscript, we have repeated this experiment and the results showed that PegIFN α indeed promoted the expression of TRIM22 protein in Hep AD38 cells. (Fig. 1C, as shown below)

C

Fig. 1. PegIFN α treatment upregulates TRIM22 expression

5. Although 107 patient samples were used for this study, the results in the tables and figures are not representative of all 107 patients. In the results the authors state (lines 236-239), for example, that 100 people were used. The explanation needs to be given of the particular sampling done for certain analyses.

Response: We apologize for the confusion. Due to the limitations of ethics and

sample size, each specimen was unable to be tested simultaneously for both Asian Screening Array (ASA) gene chips and validation of various candidate gene biomarkers. In this study, we performed peripheral blood samples using ASA gene chips. Then, we verified the levels of these differentially expressed genes by qRT-PCR. Our study was limited by the variable volumes of patient sample available, and with each additional test and validation performed, remaining sample volume further decreased past usable levels, preventing us from uniformly assaying all patient samples.

6. For the RNA fold analysis, the authors showed that a nucleotide change could induce changes in the secondary structure. But the time interval of the actinomycin D test is so long that it basically reaches the plateau after 2 hours, and it is recommended to shorten the time interval.

Response: Thanks for your very thoughtful suggestion. In the revised manuscript, according to the reviewer's comment, cells were seeded in 6-well plates at a density of 2×10^5 cells per well for 24 hours, then cells were collected at 0 h, 0.5 h, 1 h, 1.5 h, 2 h and 3 h after actinomycin D treatment. The mRNA level of *TRIM22* was detected by qRT-PCR. We found that the half-life of mRNA of SNP rs10838543 CC genotype in *TRIM22* ($t_{1/2}=1.50$ h) was still significantly longer than TT genotype ($t_{1/2}=0.57$ h) (Fig. 2F, as shown below, lines 261-262)

Fig. 2. The SNP rs10838543 CC genotype in **TRIM22** has higher *TRIM22* mRNA level and more stable than TT genotype

September 8, 2023

Mr. Qishui Ou
Fujian Medical University
Fuzhou
China

Re: Spectrum02247-23R1 (A novel TRIM22 gene polymorphism promotes the response to PegIFN α therapy through cytokine-cytokine receptor interaction signaling pathway in chronic hepatitis B)

Dear Mr. Qishui Ou:

Your manuscript has been accepted, and I am forwarding it to the ASM Journals Department for publication. You will be notified when your proofs are ready to be viewed.

Sincerely,

Wei Liu
Editor, Microbiology Spectrum
